From the perspective of rumen microbiome and host metabolome, revealing the effects of feeding strategies on Jersey Cows on the Tibetan Plateau

Yuan Niuniu 1
Wang Yicui 2
Pan Qihao 1
Zhao Li 3 4
Qi Xiao 5 6
Sun Shihao 1
Suolang Quji 3 4
Ciren Luobu 3 4
Danzeng Luosang 3 4
Liu Yanxin 2
Zhang Liyang 1
Gao Tengyun 1
Basang Zhuza 3 4
Lian Hongxia lhx263@sina.com 1
Sun Yu sunyu95@163.com 1 3 4
1 Henan Agricultural University, College of Animal Science and Technology , Zhengzhou , Henan , China
2 Henan University of Traditional Chinese Medicine, College of pharmacy , Zhengzhou , Henan , China
3 Tibet Academy of Agricultural and Animal Husbandry Science, Institute of Animal Science , Lhasa , China
4 State Key Labobatory of Hulless Barley and Yak Germplasm Resources and Genetic Improvement , Lhasa , China
5 National Animal Husbandry Service , Beijing , China
6 Tibet Academy of Agricultural and Animal Husbandry Science , Lhasa , China
Flay Kate
Electronic publication date: 2023 Sep 11
Publication date: 2023
Volume: 11
Electronic Location ID: e16010
Received 2022 Nov 14; Accepted 2023 Aug 10
Copyright: ©2023 Yuan et al.
Copyright year: 2023
Copyright holder: Yuan et al.
License: This is an open access article distributed under the terms of the Creative Commons Attribution License, which permits unrestricted use, distribution, reproduction and adaptation in any medium and for any purpose provided that it is properly attributed. For attribution, the original author(s), title, publication source (PeerJ) and either DOI or URL of the article must be cited.
License URL: https://creativecommons.org/licenses/by/4.0/

Keywords: Jersey cattle, Tibet Plateau, Feeding methods, TMR, Grazing, Rumen bacteria, Rumen fermentation, Serum metabolism

Funding: Tibet Autonomous Region Xigaze City Science and Technology Planning project RKZ2021KJ12 the Project of the central government guiding local scientific and technological development XZ202201YD0007C The China Agriculture (Dairy) Research System CARS-36 This work was supported by the Tibet Autonomous Region Xigaze City Science and Technology Planning project (No. RKZ2021KJ12), the Project of the central government guiding local scientific and technological development (No. XZ202201YD0007C), and the China Agriculture (Dairy) Research System (No. CARS-36). The funders had no role in study design, data collection and analysis, decision to publish, or preparation of the manuscript.

==============================
Background

Previous studies have discussed the effects of grazing and house feeding on yaks during the cold season when forage is in short supply, but there is limited information on the effects of these feeding strategies on Jersey cows introduced to the Tibetan Plateau. The objective of this study was to use genomics and metabolomics analyses to examine changes in rumen microbiology and organism metabolism of Jersey cows with different feeding strategies.

Methods

We selected 12 Jersey cows with similar body conditions and kept them for 60 days under grazing (n = 6) and house-feeding (n = 6) conditions. At the end of the experiment, samples of rumen fluid and serum were collected from Jersey cows that had been fed using different feeding strategies. The samples were analyzed for rumen fermentation parameters, rumen bacterial communities, serum antioxidant and immunological indices, and serum metabolomics. The results of the study were examined to find appropriate feeding strategies for Jersey cows during the cold season on the Tibetan plateau.

Results

The results of rumen fermentation parameters showed that concentrations of acetic acid, propionic acid, and ammonia nitrogen in the house-feeding group (Group B) were significantly higher than in the grazing group (Group G) (P < 0.05). In terms of the rumen bacterial community 16S rRNA gene, the Chao1 index was significantly higher in Group G than in Group B (P = 0.038), while observed species, Shannon and Simpson indices were not significantly different from the above-mentioned groups (P > 0.05). Beta diversity analysis revealed no significant differences in the composition of the rumen microbiota between the two groups. Analysis of serum antioxidant and immune indices showed no significant differences in total antioxidant capacity between Group G and Group B (P > 0.05), while IL-6, Ig-M , and TNF-α were significantly higher in Group G than in Group B (P < 0.05). LC–MS metabolomics analysis of serum showed that a total of 149 major serum differential metabolites were found in Group G and Group B. The differential metabolites were enriched in the metabolic pathways of biosynthesis of amino acids, protein digestion and absorption, ABC transporters, aminoacyl-tRNA biosynthesis, mineral absorption, and biosynthesis of unsaturated fatty acids. These data suggest that the house-feeding strategy is more beneficial to improve the physiological state of Jersey cows on the Tibetan Plateau during the cold season when forages are in short supply.

Introduction

The Jersey cow, which originated in the United Kingdom, has been widely introduced in various areas including Tibet due to its strong disease resistance, rough feeding tolerance, high feeding utilization rate, and high milk fat content (which is the origin of butter) (Kong et al., 2021). Liu et al. (2022) found that Jersey cows were the species that had the best adaptation out of all the breeds that had been introduced to high-altitude regions. The Tibetan Plateau, one of the most important pastoral areas in China, has unique environmental conditions like high ultraviolet radiation, low atmospheric pressure, and low oxygen concentration. Native Tibetan yaks are well adapted to grazing on natural grasslands as a traditional feeding strategy, particularly during the cold season when forages are in short supply. Studies have shown that in terms of energy metabolism, yaks are more capable of utilizing energy sources, absorbing nutrients more effectively, and requiring less energy to sustain life (Liu et al., 2023a). However, it remains unclear whether the traditional grazing strategy on the Tibetan Plateau is beneficial to the growth and health of introduced Jersey cows during the cold season when forages are in short supply.

The different feeding strategies have a direct impact on the ruminal microbiota (Xue et al., 2017). Rumen is an important digestive organ for ruminants, and has a wide range of rumen microbes that are extremely sensitive to feeding strategies. The feeding strategy affects the ruminal fermentation of yaks and alters the core bacterial diversity (Dai et al., 2022) and also has a considerable impact on the fecal microbiota at the phylum and family levels of Hanwoo steers (Jung et al., 2022). There were significant differences in bacterial composition and the concentration of ruminal VFAs (especially propionate and butyrate) in Tibetan Gangba sheep fed by different feeding strategies (Jize et al., 2022). These studies have found that in both house-feeding and grazing strategies, house-feeding improved rumen development and increased the production of yak rumen acetate, propionate, and microbial protein, but the rumen bacterial and microbial richness was lower in the house-feeding yaks than in grazing yaks (Xu et al., 2021; Huang et al., 2021; Fang et al., 2015). In addition, it has been shown that feeding strategy has an effect on serum metabolism, antioxidant capacity, and immunity of cattle (Zhang, Wang & Guo, 2021; Jung et al., 2022). Polyphenolic compounds in grazing forage has been found to enhance the immunity capacity and inflammatory response in yaks, as well as improve their antioxidant capacity (Cui et al., 2016a). Serum metabolomics can be used to measure variations in all metabolites produced by organisms as their nutritional condition changes, and it can also directly show the information of the metabolic state (Noguchi, Sakai & Kimura, 2003; Gil-Solsona et al., 2017). Research has shown that grazing with supplementary feeding changed the serum metabolite levels of yak heifers and improved the metabolic intensity of lipids and proteins (Xue et al., 2021; Zhou et al., 2020).

With the rapid development of biotechnology, it is now possible to use 16S rRNA and metabolomic analyses to study differences in Jersey cows fed by different feeding strategies. Specifically, the composition of the bacterial community in the rumen can be examined through 16S rRNA gene amplicon sequences (Pu et al., 2023). Metabolomics based on mass spectrometry (MS) is a method to study the overall changes of small-molecule metabolites, reflecting the physiological activities in organisms (Kong et al., 2021). Therefore, we used 16S rRNA and metabolomics to study which feeding strategy is appropriate for Jersey cows in Tibetan Plateau during the cold season when forage resources are in short supply. The results of this study could shed important light on the best ways to feed Jersey cows in Tibetan Plateau. According to our hypothesis, different feeding strategies could influence microbiota and fermentation in the rumen, as well as the metabolites in the serum.

Materials & Methods

Animals and experimental design

All animal experimental procedures were approved by the Animal Care and Use Committee of Henan Agricultural University (Approval number: HENAU-2021-025). The experiment was conducted at the Jersey Cattle Breeding base of Shigatse, Tibet (3,890 m above sea level), where Jersey cows were introduced from the same batch and kept at the breeding base for more than 2 years. At this experimental base, two feeding strategies have been adopted: grazing and house-feeding. The grazing cows were grazed on fixed pasture and supplemented with concentrate feed. House-feeding cows were fed total mixed rations(TMRs) indoors. Six Jersey cows (weighing 419.5 ± 8.83 kg) were used in each of the two feeding strategies when the experiment began in mid-September. According to previous feeding strategies, Grazing group(Group G) was fed and supplemented with 2.5 kg/d of concentrate per head, while house-feeding group (Group B) received the TMR diet. Group G was fed concentrate at 08:00 and 19:00 daily and grazed from 09:00 to 18:00 in a fixed natural pasture. Kobresia (Cyperaceae) and Stipa (Poaceae) are the dominant species in the grazing pastures. Group B was fed TMR at 08:00, 13:00 and 19:00 daily. Group B cows can move freely in the exercise yard outside the house. Jersey cows in both groups were able to drink water freely. The chemical composition of the TMR, pasture and concentrate are presented in Data S1. After 60 days, the experiment came to an end in mid-November.

Sample collection

Test cows’ blood was collected by puncturing their tail veins with a vacuum tube without any anticoagulants and being left to stand for 30 min, and the supernatant was obtained by centrifuging the blood at 3000 × g (4 °C) for 15 min and stored at −80 °C for later etermination of antioxidant parameters, immunological parameters and metabolites.

After 2 h of feeding concentrate in group G and TMR diet in group B, three cows were randomly selected from each of group G and group B. 80 ml of ruminal fluid was taken from each cow by transoral gavage. The first 30 ml of ruminal fluid was discarded to avoid saliva contamination, and the remaining was filtered through four layers of gauze and stored in liquid nitrogen for use in future concentration assays of volatile fatty acids (VFAs) and ammonia nitrogen (NH3-N) and DNA extraction. All samples were collected on the last day of the experiment period.

Feed analysis

Samples of TMR, concentrate and forages were collected once a week and mixed for composition analysis. Forages were randomly taken from ten quadrats (50 cm × 50 cm) in the pasture. The interval between quadrats was more than 100 m and the quadrats can reflect the vegetation composition and community of this pasture. Samples of TMR, concentrate and forages were dried in an oven at 60 °C for 48 h until constant weight. They were ground with a grinder, and the nutrient content was determined by one mm screening. Crude protein (CP) is measured as described by the Association of Official Analytical Chemistry AOAC, 2005 methods. The concentration of phosphorus was determined by the photometric method and of calcium by the atomic absorption spectrophotometric method (method 985.01). Neutral detergent fiber (NDF) concentration was measured with heat stable α-amylase and sodium sulfite and expressed inclusive of residual ash content, and acid detergent fiber (ADF) concentration was determined by using fiber bags and a fiber analyzer following Van Soest, Robertson & Lewis (1991).

Ruminal fermentation parameters analysis

The frozen samples were thawed at 4 °C and centrifuged at 3000 × g at 4 °C for 10 min. The VFAs were measured by using gas chromatography (Agilent 6890N, Santa Clara, CA, USA) with a chromatographic column (HP 19091N-213), according to the method of Wang et al. (2023). The NH3-N concentration of rumen fluid was determined colorimetrically (Spectrophotometer U-2900; Hitachi, Tokyo, Japan) by following the method (Weatherburn, 1967).

DNA extraction and analysis of bacterial community in rumen

Total genomic DNA samples were extracted as previously described by Yang & Yang (2021). Specifically speaking, total genomic DNA samples were extracted by using the OMEGASoil DNAKit (M5635-02) and following the manufacturer’s instructions, as well as stored at −20 °C prior to further analysis. The quantity and quality of extracted DNAs were measured by using a NanoDrop NC2000 spectrophotometer and agarose gel electrophoresis, respectively. PCR amplification of the bacterial 16S rRNA genes V3–V4region was achieved by using the forward primer 338F (5′-ACTCCTACGGGAGGCAGCA-3′) and 806R (5′-GGACTACHVGGGTWTCTAAT-3′). PCR amplicons were quantified by using the Quant-iT PicoGreen dsDNA Assay Kit (Invitrogen, Carlsbad, CA, USA). After the quantification step, amplicons were pooled in equal amounts, and paired-end 2 × 250 bp sequencing was performed by using the Illumina MiSeq platform with MiSeq Reagent Kit v3.

Antioxidant immune index analysis of serum

Serum concentrations of tumor necrosis factor-α (TNF- α), interleukin (IL)-4, IL-6, and IL-17 were measured by using commercial ELISA kits (Meimian, China), and serum immunoglobulin (IgA, IgG, and IgM), superoxide dismutase (SOD), glutathione peroxidase (GSH-Px), malonaldehyde (MDA), and total antioxidant capacity (T-AOC) were measured by using commercial colorimetric assay kits (Meimian, China).

LC−MS metabolomics analysis of serum

The metabolomics samples were thawed at 4 °C, and 100 µL of samples was taken and added to precooled methanol/acetonitrile/aqueous solution (2:2:1, V/V), vortexed, and ultrasonicated at low temperature for 30 min, allowed to stand at −20 °C for 10 min, and centrifuged at 14000 × g for 20 min at 4 °C. The supernatant was vacuum dried, subjected to mass spectrometry analysis with 100 µL acetonitrile aqueous solution (acetonitrile: water =1:1, V/V) redissolved, vortexed, and centrifuged at 14000 × g at 4 °C for 15 min, and the supernatant was then sampled for further analysis. The samples were separated by an Agilent 1290 Infinity LC ULTRA High-Performance Liquid Chromatography (UHPLC) HILIC column. The column temperature was 25 °C, and the flow rate was 0.5 mL/min with an injection volume of 2 µL. The samples were placed in an automatic sampler at 4 °C during the whole analysis. To avoid the influence of instrument detection signal fluctuation, a random sequence was used for the continuous analysis of samples. Quality control (QC) serum samples were inserted into the sample sequence to monitor and evaluate the stability of the system and the reliability of the experimental data. An AB Triple TOF 6600 mass spectrometer was used to collect the first- and second-order spectrograms of the samples.

Statistical analysis

Ruminal fermentation parameters, serum antioxidant and immune indices were analyzed by using the SPSS version 20 software (IBM). Differences in mean were determined by the Tukey test. Statistical significance was set at P < 0.05; P < 0.01 was considered highly significant.

16S rRNA

Analysis of raw 16S rRNA gene sequence data was mainly performed by using QIIME and the R package (v3.2.0; R Core Team, 2015). Microbiome bioinformatics was performed in QIIME2 2019.4 with slight modifications. Briefly, raw sequence data were demultiplexed by using the demux plugin and then were primers-cutted with cutadapt plugin. Sequences were then quality filtered, denoised, merged and chimera removed by using the DADA2 plugin. After the above-mentioned steps, 997,885 sequences of samples were extracted, with an average of 416 bp per sequence; After removing the chimeras, 407,923 high-quality sequences were obtained, and 323,300 amplicon sequence variants were obtained after removing singletons. Species were annotated by using Greengenes database. Sequence data analyses were mainly performed by using QIIME2 and R packages (v3.2.0). ASV-level alpha diversity indices, such as Chao1 richness estimator, observed species, Shannon diversity index, Simpson index, and Good’s coverage were calculated by using the ASV table in QIIME2. Beta diversity analysis was performed to investigate the structural variation of microbial communities among groups by using Bray-Curtis metrics, visualized via principal coordinate analysis (PCoA) was also conducted based on the genus-level compositional profiles, as well as sequence data were analyzed as previously described in Zhang et al. (2023). The significance of the differentiation of microbiota structure among groups was assessed by ANOSIM (Analysis of similarities) using QIIME2. The taxonomy compositions and abundance were visualized by using MEGAN and GraPhlAn. Venn diagram was provided to visualize the shared and unique ASVs among samples or groups by using R package “Venn Diagram”, based on the occurrence of ASVs among samples/groups. The output files were further analyzed by using the Statistical Analysis (STAMP) package (Parks et al., 2014).

LC-MS metabolomics

The raw MS data for the metabolomics for peak alignment, retention time correction, and peak area extraction were analyzed by using the Compound Discoverer 3.0 program (Huan et al., 2017). Metabolite structure identification was performed by a method of accurate mass number matching (<25 ppm) and secondary spectral matching to retrieve the MZcloud database (Liu et al., 2019b).

The SIMCA-P14.1 (Umetrics) was used for the pattern recognition multivariate statistical analysis. After Pareto scaling, principal component analysis and orthogonal PLS-DA (OPLS-DA) were performed. Principal component analysis was used to visualize the data set, including the similarities and differences, while OPLS-DA was performed to maximize the covariance between the measured data and the response variable. To avoid over-fitting, model validity was evaluated by a permutation test. Metabolites with variable importance projection(VIP) >1 in multidimensional statistical analysis and P value <0.05 in univariate statistical analysis were considered significantly different, and the corresponding variable importance in the projection (VIP value) was calculated in the OPLS-DA model. The differential metabolites identified were subjected to clustering and Kyoto Encyclopedia of Genes and Genomes (KEGG) pathway enrichment analysis.

Results

Ruminal fermentation parameters

The rumen fermentation parameters in the two feeding strategies are shown in Table 1. The contents of acetic acid, propionic acid, and NH3-N in the rumen fluid of Group B were significantly higher than those of Group G (P < 0.05), while the butyric acid and acetic propionic acid ratios were not significantly different (P > 0.05).

Table 1 Rumen fermentation parameters and bacteria alpha diversity indices of Jersey cattle under two feeding strategies.

Items	Group (Mean ± SEM)	P value	
	G	B		
Acetate (mg/L)	36.76 ± 0.22	56.87 ± 7.85	0.041	
Propionate (mg/L)	24.48 ± 1.66	36.85 ± 0.28	0.011	
Butyrate (mg/L)	2.38 ± 0.41	2.67 ± 0.03	0.616	
A/P	1.52 ± 0.10	1.55 ± 0.22	0.898	
NH3-N (mg/dL)	2.94 ± 0.40	6.50 ± 0.88	0.021	
Chao1	5304.99 ± 19.85	3848.26 ± 478.97	0.038	
Observed species	3837.07 ± 50.74	2924.63 ± 400.52	0.087	
Shannon	10.46 ± 0.04	8.98 ± 1.34	0.333	
Simpson	1.00 ± 0.00	0.93 ± 0.07	0.374	
Notes.

G Grazing group

B TMR house feeding group

Ruminal bacteria analysis

The Good-coverage index revealed that the assay covered more than 95% of samples, indicating that the sequencing data were able to accurately cover all species in the samples. As seen from the sparse curve, the number of species detected gradually rose to a peak, indicating that the sequencing results were sufficient to reflect the diversity contained in the current sample for further analysis. As seen from the Venn diagram (Figs. 1A–1B), a total of 1978 common ASVs were identified in both groups, with 6621 ASVs and 4848 ASVs identified in Group G and Group B, respectively. Table 1 shows the alpha index of the rumen bacterial community in the two groups. The Chao1 index in Group G was significantly higher than that in Group B (P = 0.038), while the observed species, Shannon, and Simpson indices were not significantly different between the two groups (P > 0.05). The PCoA plots (Fig. 1C) were further analyzed for differences in bacterial community between Group G and Group B. The beta diversity analysis did not respond to significant distances between the two groups, and adnoism analysis (Fig. 1D) showed no significant differences in the rumen bacterial community between Group G and Group B (R = 0.55, P = 0.08).

Figure 1 Venn diagram, ASV rank abundance curve, beta community diversity analysis of rumen microbiota under two feeding strategies.

(A) Venn diagram showing operational taxonomic units shared between the two groups. (B) ASV rank abundance curve. (C) Principal component analysis (PCoA) of the bacterial community of Jersey cows under two feeding strategies. PCo1, 1st principal component, PCo2, 2nd principal component. The percentage of variation explained by each principal coordinate is indicated on the axes. (D) A boxplot of the distribution of distances between samples. G, Grazing group; B, TMR house feeding group.

By species composition analysis, Firmicutes and Bacteroidetes were the main phyla at the phylum level, accounting for 59.11% and 37.19% in Group G and 62.56% and 31.23% in Group B respectively. At the genus level, Group G and Group B were dominated by Ruminococcus (3.11%, 4.69%), Succiniclasticum (2.95%, 2.64%), Prevotella (2.51%, 2.66%), Butyrivibrio (1.73%, 1.79%), and Clostridiaceae_Clostridium (1.42%, 2.08%), and Fusobacterium accounted for 15.65% in Group B and none in Group G.

At the phylum level, the relative abundances of the phyla TM7 and Fusobacteria were significantly higher in Group B than in Group G (P < 0.05), while there was no significant difference in the phyla Bacteroidetes, Firmicutes, and Proteobacteria (P > 0.05). At the genus level, the relative abundances of the genera Ruminococcus, Anaerostipes and Fusobacterium were significantly higher in Group B than in Group G (P < 0.05), while there were no significant differences in the genera Succiniclasticum, Prevotella, and Coprococcus (P > 0.05) (Figs. 2A and 2B).

Figure 2 Effects of feeding strategies on rumen microbiota in Jersey cows.

(A) The bacteria with significant differences between two feeding strategies at the phylum level. (B) The bacteria with significant differences between the two feeding strategies at the genus level. G, Grazing group; B, TMR house feeding group; an asterisk (*) indicates P < 0.05.

Serum antioxidant and immune

The effects of different feeding strategies on serum antioxidant and immune indicators of Jersey cows are shown in Table 2. There were no significant differences in antioxidant indicators between the two groups (P > 0.05). The Ig-M, IL-6, and TNF-α levels in Group G were significantly higher than those in Group B (P < 0.05), while the rest of the serum indicators did not differ significantly between the two groups (P > 0.05).

Table 2 Serum antioxidant capacities and immunity indices of Jersey cattle under two feeding strategies.

Items	Group (Mean ± SEM)	P value	
	G	B		
T-AOC (µmol Trolox/ml)	5.88 ± 0.03	5.87 ± 0.03	0.782	
MDA (nmol/ml)	1.97 ± 0.11	1.72 ± 0.06	0.106	
SOD (U/ml)	68.45 ± 8.17	78.22 ± 10.40	0.466	
GSH-Px (nmol/min/ml)	190.76 ± 7.08	190.18 ± 7.43	0.957	
IL-4 (ng/L)	76.72 ± 0.79	75.52 ± 1.36	0.424	
IL-6 (ng/L)	18.29 ± 0.33	17.06 ± 0.45	0.039	
IL-17 (ng/L)	62.74 ± 1.18	65.53 ± 2.24	0.243	
IgA (µg/ml)	176.09 ± 2.62	185.12 ± 4.11	0.067	
IgG (µg/ml)	1901.27 ± 37.87	1899.54 ± 66.26	0.981	
IgM (µg/ml)	107.95 ± 1.94	98.01 ± 2.39	0.005	
TNF- α (ng/L)	298.76 ± 2.68	271.01 ± 9.07	0.003	
Notes.

G Grazing group

B TMR house feeding group

Metabolomics of serum samples

Positive and negative ionization modes by using PCA demonstrated a major, unsupervised separation between the two groups (Fig. 3A). The multivariate analysis was performed by OPLS-DA with positive and negative ionization modes to differentiate the groups more clearly (Figs. 3B and 3C). The samples in the score plots were all within the 95% Hotelling T2 ellipse, demonstrating the validity of the OPLS-DA model, and the two groups were clearly distinguished from each other. In the positive and negative ionization modes, 149 major serum differential metabolites (Table S2) were identified between the two groups based on VIP >1.0, P < 0.05, of which 54 lipids and lipid-like molecules were the major metabolites, along with 41 organic acids and derivatives, 21 organoheterocyclic compounds, 20 benzenoids, 6 organic oxygen compounds, three alkaloids and derivatives, two organic nitrogen compounds, 1 nucleoside/nucleotide and analog, and 1 phenylpropanoid and polyketide.

Figure 3 Principal component analysis (PCA) model score scatter plot, orthogonal partial least-squares discriminant analysis (OPLS-DA) model, and permutation test of serum metabolic profiling.

(A) Principal component analysis (PCA) in positive (Pos, R2X = 0.525) and negative (Neg, R2X = 0.550) ion modes. (B–C) OPLS-DA models of positive (R2Y = 0.994, Q2 = 0.813) and negative (R2Y = 0.990, Q2 = 0.918) mode ionization. G, grazing group; B, TMR house feeding group.

Figure 4 shows the heatmap of the cluster analysis of the differential metabolites screened by VIP > 1, P < 0.05 in positive and negative ionization modes in Group G and Group B. The cluster analysis allowed visualization of the differences in serum metabolites between the two groups. Alloxanthin, hypoxanthine, serotonin, cholic acid, cytosine, lactose, niacinamide, uracil, D-glutamine, glycine, uridine, adenine, phenaceturic acid, guanidinosuccinic acid, and lithocholylglycine were significantly up-regulated in Group B (FC >1, P < 0.05), and phenylalanine, DL-isoleucine, DL-threonine, phenanthrene, isoleucine, 3-hydroxybutyric acid, linoleic acid and oleic acid were significantly up-regulated in Group G (FC < 1, P < 0.05).

Figure 4 Hierarchical cluster analysis and heat map of serum metabolites under different feeding strategies.

G, grazing group; B, TMR house feeding group.

KEGG enrichment analysis was performed according to the identified serum differential metabolites. Figure 5 shows the top 20 significantly enriched metabolic pathways under the two different feeding strategies, with serum differential metabolites mainly enriched in the biosynthesis of amino acids, protein digestion and absorption, ABC transporters, aminoacyl-tRNA biosynthesis, mineral absorption, and biosynthesis of unsaturated fatty acids.

Figure 5 KEGG pathway enrichment map.

The horizontal coordinate indicates the number of differential metabolites contained in the pathway, the vertical coordinate refers to the name of the metabolic pathway, and the values in the graph represent the P-value for the enrichment analysis.

Discussion

Ruminal fermentation parameters

The rumen is the largest and primary organ for the digestion and fermentation of feed by symbiotic microorganisms in ruminant animals, which results in the generation of volatile fatty acids (Hess et al., 2011). Ruminal VFAs mainly include acetic, propionic, butyric acids, and other short-chain fatty acids, of which over 75% of ruminal VFAs are absorbed through the rumen epithelium and utilized as energy resources (Bergman, 1990). This suggested that greater energy intake and nutrition may be available to support the growth of ruminant animals when VFAs are absorbed and converted to ruminal nutrients (Jize et al., 2022). In the current study, the levels of acetic acid, propionic acid, and NH3-N in Group B were significantly higher than those in Group G, which is consistent with the results of studies on TMR feeding and grazing-supplemented concentrate feeding of Holstein cows (Hartwiger et al., 2018). The same results were found in Friesian cows fed TMR diets and a low-roughage diet (Sutton et al., 2003). Ruminal ammoniacal nitrogen is obtained mainly from the protein degraded by the rumen (Felisberto et al., 2011). Ruminal ammonia-N is the major N source for the growth of microbial bacteria. The concentration of rumen NH3-N in Group B was higher than in Group G, indicating a higher microbial protein synthesis in Group B. Rumen fermentation results showed that grazing provides Jersey cows with a lower amount of nutrients than from house-feeding diets.

Ruminal microbiota

The Chao1 index of Group G was significantly higher than that of Group B, while there was no difference in the Shannon and Simpson indices. The greater the Chao1 and observed species indices of the alpha index, the greater the richness of the community (Chao & Shen, 2004). The higher the Shannon and Simpson index values are, the greater the diversity of the community (Simpson, 1949; Shannon et al., 2003). This study showed that the richness of the rumen bacterial community in Group G was higher than that in Group B. The greater richness was a consequence of the high number of plant species in Group G when compared with Group B. The increased rumen microorganism richness in Group G reflects its higher ability to use high-fiber pasture. This is similar to the results of previous studies: the richness of the rumen bacterial community was higher in grazing yaks than in house-feeding yaks, and increasing the proportion of concentrate in the ration significantly reduced the rumen bacterial diversity in yaks (Kong et al., 2010; Huang et al., 2021; Liu et al., 2019a). Similar results were observed in yaks and Tibetan sheep, and the rumen richness in the house-feeding group was significantly lower than that in the grazing group (Xue et al., 2017). Beta diversity analysis showed that the two feeding strategies did not significantly alter the composition of the rumen microbiota.

The dominant rumen bacterial community at the phylum level in both feeding strategies were Firmicutes and Bacteroidetes, with most bacteria in Firmicutes degrading cellulose and those in Bacteroidetes improving carbohydrate utilization, which is consistent with the findings of the dominant phyla in highland yaks (Pope et al., 2012). The dominant phyla were the same in different feeding strategies. Similar to previous findings, the dominant phyla do not change as fiber levels in the diet change (Xue et al., 2016). TM7 and Fusobacteria were significantly higher in Group B than in Group G. Most of the bacteria in Fusobacteria were conditionally pathogenic and could lead to certain diseases under certain circumstances. TM7 was associated with a defective mucus barrier in the intestine (Jakobsson et al., 2015). Fusobacterial and TM7 can cause certain intestine diseases in cattle, which indicates that Group B has the risk of causing intestine damage.

At the genus level, the dominant genera in both groups were Ruminococcus, Succiniclasticum, and Prevotella, with Prevotella degrading starch and protein and Ruminococcus and Succiniclasticum being closely associated with the fermentation of fibrous feeds(Kovatcheva-Datchary et al., 2015; Koike & Kobayashi, 2009). The relative abundances of Ruminococcus, Anaerostipes, and Fusobacterium were significantly higher in Group B than in Group G. Anaerostipes can produce SCFAs (e.g., propionate and butyrate) from sugar (Thi Phuong Nam et al., 2021). Ruminococcus and Fusobacterium are known cellulose-degrading bacterial species that produce a variety of enzymes to degrade cellulose to acetic acid (Louis & Flint, 2009). The results have shown that TMR diets in Group B can promote the growth and multiplication of Ruminococcus, Anaerostipes, and Fusobacterium in the rumen, thereby producing more VFAs, improving the fermentation environment of the rumen, and promoting the absorption and utilization of nutrients by the organism.

Serum antioxidant and immunity index

In highland areas, factors such as low oxygen, strong UV exposure, and drastic temperature changes lead to increased levels of oxidative stress in the body, which can be exacerbated by nutritional deficiencies (Jefferson et al., 2004). Oxidative stress is a mechanism that destroys cellular molecules and produces inflammatory cytokines that further affect the body’s capacity to produce various physiological responses (Sordillo, 2013; Spears & Weiss, 2008). Therefore, antioxidant capacity is essential for the health of the organism. When assessing the antioxidant capacity of animals, serum SOD and T-AOC concentrations were used to reflect the body’s antioxidant defense system against free radical metabolites (Rajapakse et al., 2005). MDA is one of the end products of membrane lipid peroxidation, and its level can be used as an indicator of the severity of cellular stress. GSH-Px, an important antioxidant enzyme, protects against the harmful effects of oxidative stress (Gong & Xiao, 2016). Studies have shown that the antioxidant capacity of natural forages increases with altitude and grazing animals can increase their antioxidant capacity through taking forages. In this study, there was no significant difference between the serum antioxidant index of Jersey cows in Group G and Group B, indicating that there was no significant effect of roughage in pasture on the serum antioxidant index of Jersey cows. It may be related to the fact that Jersey cows were fed the same concentrate in the two feeding strategies. Studies have shown that supplemental feeding of concentrates can significantly increase the antioxidant capacity in the serum of grazing yaks (Cui et al., 2016b; Cui et al., 2016a; Zhang et al., 2020). Trace minerals have important roles in immunological functions, antioxidant capacity, and overall health in livestock (Sordillo, 2013; Spears & Weiss, 2008).

Serum IgA, IgG and IgM are immunoglobulins with antiviral and antibacterial properties (Bianco et al., 2014). Interleukins mediate T- and B-cell activation, proliferation and differentiation and play an important role in transmitting information and activating and regulating immune cells. The serum levels of IL-6, TNF-α, and IgM were significantly higher in Group G of Jersey cows than in Group B. TNF-α and IL-6 are important proinflammatory cytokines in immune stress, and IL-6 is a cytokine that induces B-cell differentiation to immunoglobulin-producing cells and regulates the biosynthesis of acute-phase proteins (Muraguchi et al., 1988). Studies on yak calves showed that with reductions in TNF-α and IL-6 in cattle, the immune capacity and overall performance were reduced (Wei et al., 2021). The increase in proinflammatory factors in the serum of Group G of Jersey cows promotes the production of immunoglobulin IgM by the body’s immune system, which makes the immune system more active and thus enhances the body’s capacity to resist disease.

Serum metabolites

Serum metabolites reflect the metabolic and health status of livestock (Liu et al., 2019a).

Serum metabolomics analysis was used to provide a new perspective for exploring nutrient utilization in Jersey cows. We also predicted the involved pathways affected by feeding strategies by using KEGG analyses.

The relative levels of betaine, hippuric acid, glutamine, cholic acid, cytosine, uracil and adenine, in the serum of Group B of Jersey cows were significantly higher than those in Group G, and the results were similar to the changes in metabolites in the serum of yaks being fed with supplements during the cold season (Xue et al., 2021). Betaine is a metabolite of choline metabolism; its main physiological role is as an organic osmoprotectant or as a methyl donor through transmethylation, and it participates in protein and lipid metabolism (Alberti et al., 2009). Betaine can promote fatty acid β-oxidation and reduce the synthesis ability of fatty acids and triglycerides, resulting in a decrease in lipid synthesis (Zeisel & Da Costa, 2009). Studies have shown that betaine in bovine serum can reduce serum urea nitrogen, increase the concentration of total protein and improve the utilization rate of nitrogen. Betaine in serum has several anti-inflammatory effects, including the inhibition of NF-κβ (Huang et al., 2007; Eom et al., 2022). Hippuric acid is derived from dietary protein degradation and can be resynthesized from aromatic compounds or cyclic sugar-type compounds by the gastrointestinal microbiota through the shikimate pathway (Lees et al., 2013). In previous studies, higher serum hippuric acid levels might also indicate more energy supplied by glucose metabolism and hormone regulation (Liu et al., 2022). Glutamine is involved in the intracellular biosynthesis of purines, pyrimidines, and nucleotides, and glutamine also has an important regulatory function in increasing protein synthesis and reducing protein degradation in skeletal muscle. A previous study revealed that compared with dairy cows, serum glutamine levels in yaks were significantly increased, which may be a metabolic adaptation to hypoxia, promoting amino acid metabolism to meet their own energy requirements. The results of this study are consistent with previous results (Lapierre et al., 2012; Huang et al., 2022; Horscroft & Murray, 2014). Cholic acid is the primary bile acid produced from cholesterol. Cholic acid combines with glycine to form glycocholic acid, which is secreted with bile into the small intestine and contributes to the absorption of nutrients. The metabolites that were significantly up-regulated in the serum of the Jersey cows in Group B were mainly related to amino acid metabolism and glucose metabolism. Group B increased their energy supply by increasing amino acid metabolism and glucose metabolism to meet their energy requirements. This was also a metabolic adaptation to the hypoxic environment, as well as improving the body’s utilization of nitrogen and increasing its absorption of nutrients.

The relative concentrations of oleic acid, linoleic acid, shikimate, 3-hydroxybutyric acid, phenylalanine, stachydrine, and alpha-tocopherol in the serum of Jersey cows in Group G were significantly higher than those in Group B. Oleic and linoleic acids are unsaturated fatty acids and the main free fatty acids in the blood for energy storage and utilization by tissues (Hocquette, Graulet & Olivecrona, 1998). The concentration of free fatty acids in the blood also reflects changes in the metabolic turnover of adipose tissue. In addition, oleic acid can be involved in a variety of vital activities, such as cell proliferation, inflammatory response, and hormone regulation, and can induce the release of inflammatory factors, such as IL-6 and TNF-α, from adipocytes, which may explain the high serum levels of IL-6, TNF- α, and Ig-M (Frommer et al., 2015; Steffen et al., 2018). 3-Hydroxybutyric acid is synthesized from the acetyl-coenzyme acetyl-CoA produced by the degradation of fatty acids and is also a marker of lipid metabolism in the body. A high serum concentration of 3-hydroxybutyric acid indicates higher levels of lipolysis (Gaertner et al., 2019). It is known that, the body promotes catabolism to provide ATP when energy intake is restricted, such as fatty acid oxidation and protein catabolism, while inhibiting ATP-consuming anabolism, such as fatty acids and proteins (Zhou et al., 2020). Shikimate undergoes biosynthesis with phenylalanine, tyrosine and tryptophan to produce alpha-tocopherol. Study results suggest that dairy cows adapt to high-altitude hypoxia by up-regulating phenylalanine metabolism and phenylalanine, tyrosine, and tryptophan biosynthetic signaling pathways (Kong et al., 2021). Alpha-tocopherol, as a component of the general plasma lipoprotein system, is transported in the blood and is the main carrier of alpha-tocopherol in plasma. Plasma lipoprotein concentration is a function of the global dynamics of lipid metabolism, and alpha-tocopherol usually modulates changes in lipid transport by cholesterol concentration (Weiss & Wyatt, 2003). Increased lipid metabolism led the Jersey cows in Group G to adapt to lipid transport by increasing the serum concentrations of tocopherols. Stachydrine is mainly found in tubers, alfalfa (Medicago sativa) and some Compositae, legumes, and other herbs. Stachydrine can significantly inhibit the mRNA levels of lipase in the liver and adipose tissue and restore the balance in the endoplasmic reticulum in the liver and adipose tissue (Trinchant et al., 2004; Lee et al., 2022). Our results showed that due to the lack of energy in Group G, the Jersey cows in Group B provided ATP by promoting fatty acid oxidation and protein hydrolysis metabolism, leading to elevated levels of some fatty acids and amino acids in the serum.

Figure 6 Summary of rumen fermentation indexes, serum antioxidant immune indexes and summary of the identified differential metabolites and their metabolic pathways.

Yellow-colored symbols represent significantly higher metabolites in group B compared with the group G, blue-colored symbols indicate lower, white-colored symbols mean no difference.

Conclusions

In conclusion, our results show that grazing provides fewer nutrients to Jersey cows than house feeding, but the immune system of Jersey cows in Group G is more active and the rumen bacterial community is more abundant than in Group G. Group B exhibited a higher abundance of bacterial genus associated with rumen fermentation, as compared to Group G. Group B increased energy supply by increasing amino acid metabolism and glucose metabolism to meet their energy requirements. Group G provided insufficient energy and provided ATP by promoting fatty acid oxidation and protein hydrolysis metabolism (Fig. 6). These data suggest that house feeding is more beneficial to the improvement of the physiological state of Jersey cows in the high-altitude areas fed by two feeding strategies during the cold season when forages are in short supply, which provides the theoretical basis for the nutritional physiology of Jersey cows on the Tibetan Plateau.

Supplemental Information

Data S1 Nutrient composition of TMR and concentrate (DM basis) %

Click here for additional data file.

Supplemental Information 2 Author Checklist—Full

Click here for additional data file.

The authors would like to thank the staff of the Cattle Breeding base of Tibet Academy of Institute of Animal Science, Tibet Academy of Agricultural and Animal Husbandry Science for their help in sample collection.

Additional Information and Declarations

Competing Interests

Author Contributions

Animal Ethics

DNA Deposition

Data Availability

The authors declare there are no competing interests.

Niuniu Yuan conceived and designed the experiments, performed the experiments, analyzed the data, prepared figures and/or tables, authored or reviewed drafts of the article, and approved the final draft.

Yicui Wang performed the experiments, analyzed the data, prepared figures and/or tables, and approved the final draft.

Qihao Pan performed the experiments, prepared figures and/or tables, and approved the final draft.

Li Zhao performed the experiments, prepared figures and/or tables, and approved the final draft.

Xiao Qi performed the experiments, analyzed the data, prepared figures and/or tables, and approved the final draft.

Shihao Sun performed the experiments, analyzed the data, prepared figures and/or tables, and approved the final draft.

Quji Suolang performed the experiments, prepared figures and/or tables, and approved the final draft.

Luobu Ciren performed the experiments, prepared figures and/or tables, and approved the final draft.

Luosang Danzeng performed the experiments, prepared figures and/or tables, and approved the final draft.

Yanxin Liu performed the experiments, analyzed the data, prepared figures and/or tables, and approved the final draft.

Liyang Zhang performed the experiments, analyzed the data, prepared figures and/or tables, and approved the final draft.

Tengyun Gao performed the experiments, analyzed the data, prepared figures and/or tables, and approved the final draft.

Zhuza Basang performed the experiments, prepared figures and/or tables, and approved the final draft.

Hongxia Lian performed the experiments, analyzed the data, prepared figures and/or tables, and approved the final draft.

Yu Sun conceived and designed the experiments, performed the experiments, analyzed the data, prepared figures and/or tables, authored or reviewed drafts of the article, and approved the final draft.

The following information was supplied relating to ethical approvals (i.e., approving body and any reference numbers):

Animal care and use procedures were approved by the Henan Agricultural University (HENAU-2021-025). All animal procedures were conducted in strict accordance with the rules and guidelines outlined by the Henan Agricultural University Animal Welfare Committee.

The following information was supplied regarding the deposition of DNA sequences:

The sequences are available at the National Library of Medicine: SRR6324372.

The following information was supplied regarding data availability:

The raw measurements are available in the Supplementary File.

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
