# Peer review of "From the perspective of rumen microbiome and host metabolome, revealing the effects of feeding strategies on Jersey Cows on the Tibetan Plateau"

_PeerJ, doi:10.7717/peerj.16010_

## Round 0.1 · original submission · Major Revisions

The reviewers have provided some very useful feedback that must be addressed. Please consider this feedback carefully, particularly focusing on the comments around the methodology and statistical analyses.

Ensure you explain, point-by-point the details of the revisions to your manuscript and your responses to the reviewers. If you are unable to address certain points, please include an explanation/justification in your rebuttal.

In addition to the reviewer comments, please find my editorial comments below:

General comments:
- Throughout – need to add a space before the brackets where there are in-text citations
- Throughout – clarify term ‘grazing supplementary feeding’ – what is meant by this? The term is used a lot (in the introduction too) but a reader may not be familiar with this. Consider re-phrasing to ‘grazing with supplementary feeding’, or something similar.
- Switching between terminology TMR, barn feeding and house feeding – update throughout manuscript (or clarify differences) to avoid confusion for the reader
- Throughout – table and figure headings to be updated to be more comprehensive – should be able to understand in isolation from main body of text.

Abstract:
- Line 33 – delete ‘in this study’.
- Methods – need to provide further detail on the management of the cattle and the different feeding systems
Introduction:
- Line 65 – be specific, which ‘animals’?
- Line 67 – what is meant by ‘reasonable’ – suggest re-phrase to emphasise the importance of optimising grazing management
- Line 68 – start new paragraph with yaks content and re-word to “Yaks are an important livestock in QTL, as they provide….”
- Line 73 – reword to remove the phrase ‘a study by’ as this is redundant info
- Line 75 – can you quantify this for the reader i.e., approximately how many Jersey cattle are in the QTL?
- Line 81 – delete “rumen microbes are” sentence – a repeat of info earlier in the paragraph
- Lines 82 – 83 – mention studies, but only a single study is cited – update
- Switching between terminology TMR, barn feeding and house feeding – update throughout so is consistent
- Line 89 – what organism(s)? Clarify if this is yak or cattle (or both)?
- Line 99 – suggest change to describe rather than reveal
- Line 101 – 102 – the hypothesis needs rephrasing for clarity

Materials & Methods:
- Must be improved. The animals and experimental design section & sample collection section need more detail, specifically:
o When was the experiment done? (Actual dates)
o Sample size calculations &/or justification for group sizes
o What was the diet before the experiment started?
o How long were the 2 different groups fed the different diets?
o The composition of the concentrate & TMR diets is given in supplemental material – the label in this table needs to be improved – should be able to be understood in isolation.
o Clarify the volumes/amount of diet offered?
o For the grassland grazing group – what forage was available? How were they managed? What area was available for grazing? Did they stay in the same area etc? This is currently very vague.
 Were samples taken for pasture analysis?
o Clarify what happened to Group G overnight
o When were the feed samples taken? How many feed samples were taken? Is the info in the table from a single sample or is it a representation of multiple samples?
o When were the blood samples collected? i.e., how long after the start of the experiment – how long had the cattle been on the respective diets? Clarify all samples were collected on the same day.
o Similarly, with the rumen fluid – when was this done? As above?
 Rationale as to sample size used here is missing.
o Line 151 – what is an appropriate amount?
o Line 161 – change in tense – should be was, rather than is
o In the statistical analysis section, rather than just the stats packages used, please provide additional information as to how the models were built, validity checked, and then interpreted. Some of this information is in the results – it should be here instead. Suggest:
 Break the stats down into subheadings for each of the parameters looked at
 The first part of the ruminal bacterial analysis results can go into the materials & methods – i.e., sequences, diversity etc.
 The PCA needs to be better described in the stats section
o Your animal/research ethics approvals and numbers need to be included here – please add statement to methods
Results:
- See comments on ruminal bacteria analysis (above)
- Line 202 & Figure 1 – define OTU on first use in text
- Lines 217-218 – this is info that should be in the materials & methods section
- Lines 235-236 – this info belongs in the materials & methods

Discussion:
Very hard to follow in the current format. The discussion needs some changes for it to be clear for the reader:
- In general, the clarity of writing is poorer in the discussion than elsewhere in the manuscript. Please revise this so that the style is the same as for the rest of the document.
- Throughout the discussion, it needs to be clear when you’re discussing your results vs when you are comparing to other literature – at the moment it is confusing for the reader – please check and adjust as needed throughout -> this is really important please and must be done
- The discussion needs to be split into shorter paragraphs – excessively long paragraphs at present make it hard to follow
- Some of the discussion is speculation – in parts, this is OK, but it needs to be clear to the reader when you are speculating based on results of other studies e.g., for some of the pathways and reasons for why you observed certain changes – please update throughout for clarity
- Line 269 – use of animals is out of context here – consider ruminants or cattle (as most appropriate)
- Line 273-274 doesn’t make sense – uses different tense – plus needs a reference.
- Line 278 – ‘a low’ rather than ‘the low’
- Line 285 – 287 is speculation rather than known ‘fact’ – please adjust so that it is presented more appropriately
- Lines 282 – 287 – blurring between other studies and your results – please make it clear for the reader what you found and how this compared to existing literature
- Line 289 – this sentence is not needed – delete
Conclusion

- Lines 456-459 appear to be speculative, rather than based on actual results of this study?

·

Basic reporting

English need to improve

Experimental design

no comment

Validity of the findings

no comment

Additional comments

The article investigates that differences in rumen fermentation, serum indicators, and serum metabolites in Jersey cattle under two different production systems on the Tibetan plateau. After reviewing this manuscript, there are some changes that need to be made in regards to this study.

1. Line 112 Is the grazing pasture fixed in one area or does it change from day to day?
2. Line 132-134 The article describes the determination of crude fat, but this is not indicated in Table S1, the schedule shows the content of calcium and phosphorus, and the method describes the test method, please check and add the analytical method.
3. Line 256-260 Metabolomics is a relative comparison between two groups, the FC value for GroupB is >1, then for GroupG the FC value should be <1, please modify accordingly.
4. Line 322 The article speculates that the forage intake of the grazing group was lower than that of the housed group, but the trial did not measure the intake and there are no relevant data to prove this, so it cannot be discussed based on speculation.
5. Line 407 There are no energy levels in the nutrient composition of the diets and concentrates, the article describes higher levels of concentrates than TMR diets, please double-check the table for missing data.
6. Line 407 There is no data to suggest that Jersey cattle rely mainly on concentrate for energy, please revise the discussion.

Reviewer 2 ·

Basic reporting

Generally well written. A few grammatical issues such as:
"a Total Mixed Ration" rather than "Total Mixed Ration"
"Serum concentration" rather than "Serum concentrations"

It would be good to present the proportions of the volatile fatty acids and also the total volatile fatty acid concentration. It appears that the Group B produced more VFA and so it would be good to know if the proportions were the same our if it was just a total production effect on acetic and propionic acid production.

In Figure 2, don't need parts a) and c). Its a bit repetitive

Some of the Tables e.g., Tables 1-4, could be combined to save space.

The discussion needs to allude to the biological and production importance of the differences found in serum metabolites?

Experimental design

This is good. Potentially low numbers of animals but 6 per treatment should provide enough statistical power for the attributes measured.

Validity of the findings

This is fine

Reviewer 3 ·

Basic reporting

It is interesting to read this study about the effect of housing-feeding systems on rumen microbiota and serum metabolism of Jersey cattle at Qinghai-Tibet Plateau. However, the way that the authors wrote is a bit lack logistics, and the flow of the whole story is not clear, the English written is also ambiguous and not professional. Altogether, it is difficult for me to follow and understand what the authors wanted to express.

Experimental design

For the experimental design, I did not get the point why the authors only randomly collected rumen fluid from three cattle as the sample size for the whole study is already small (n=6 per group). Is the dataset enough represented for the differences observed in rumen microbiota? In terms of the method part, the bioinformatic analysis of rumen microbiota is missing, and the description for statistical analysis is not clear.

Validity of the findings

When finishing the reading, I did not get what kind of research questions the authors wanted to address and have a strong feeling that the dataset is not enough to support the hypothesis that the authors speculated.

Additional comments

My comments are enclosed below point by point:

Title for manuscript: How to define “the perspective of genomic”? I did not get the point about which part of your study is related to genomics.

Abstract:
The description for the background is not logistical, why there were relevant studies on yak, then the authors wanted to study this feeding system on Jersey cattle? And no information was included why you wanted to study from the perspectives of genomics and metabolomics.
Line 44-45: From my understanding, principal component analysis (PCA) is just a method to visualize your data, can you say PCA is not significant without mentioning statistical methods?
Line 51: The differential metabolites were mainly enriched in the….You said “mainly” here, but why did you list so many different metabolites and other metabolic pathways? The structure of this sentence should be restructured.

Introduction:
In general, the induction section should be restructured as it is difficult to follow.

Taking the first paragraph as an example, I did not get the points what authors wanted to do and address when finish the reading. The first sentence is normally the topic sentence of a paragraph, so I think it should be better to start with the induction for the current conditions of Jersey cattle (your subject of this study) in Qinghai-Tibetan Plateau, instead of beginning with the introduction for Qinghai-Tibetan Plateau.
In the second paragraph, the authors introduced differences in feeding systems can impact the rumen microbiota and serum metabolites in yak, is it beneficial for ruminant productivity or health? So authors wanted to study the effect of feeding systems on Jersey cattle? In other words, what are the research questions that you want to answer and solve?

Materials and Method
Line 110: What does TMR mean?
Line 123: The reason for selecting the number of cows should be pointed out, why the authors decide to only choose three cattle instead of the whole group (six cows) as the sample size was already small?
Line 135: What does DNA analysis mean? I think the authors wanted to say, “microbiota profiling”?
Line 137-142: The method used here is not clear for me and we cannot repeat it according to the descriptions. Some detailed questions below:
Which method did the authors use for the total DNA extraction? What is the detailed protocol and materials for DNA amplification? Did the authors include the controls (e.g., blank samples or mock communities) to evaluate the quality of sequencing or check the potential contamination?
Line 161: What does QC mean?

Statistical analysis
Line 165-186: 1) The description of bioinformatic analysis for rumen microbiota is missing, the authors are required to describe more in detail, like which version of database did authors use? 2)The statistical analysis either for the relative abundance of rumen microbiota or metabolites, which method did authors use to adjust the p-value?

Line 166 -167: It is not clear for me about the mixed procedure the authors used? It is better to describe more in detail.
Line 169: Instead of OTU clustering, Amplicon sequence variant (ASV) is recommended, which has a significant advantage to more precise identification of microbes?
Line 170: Which UniFrac distance metric did the authors use?

Results
Line 197: amplicon sequence variants or OTUs?
Line 207: Did the authors perform PCA or PCOA? From my understanding, PCoA should be performed if the authors used UniFrac distance metric.
Line 217-224: which test did authors perform for the statistical analysis? was the p-value adjusted to avoid the random fluctuation? The same questions for the other parameters the authors measured.
In addition, the quality of figures should be improved. The texts are even not readable in some figures.

Discussion
Line 273-274: References are required for this sentence.
Line 289-290: This expression of this sentence is too ambiguous.
Line 301-302: Which level did the authors use to perform beta diversity analysis? ASV or genus level?
Line 302: There are differences about the use for the term of microbiota, microbiome and flora, better not to mixed up.
Line 305-306: This sentence here is alone, so the connection word is needed for the transition.
Line 314-316: What is the purpose of this sentence here?
Line 318-319: Suggest the authors to change the sentence “Anaerostipes produces SCFAs, such as propionate and butyrate, from sugar (Nam et al., 2021)” to “Anaerostipes can produce SCFAs (e.g., propionate and butyrate) from sugar (Nam et al., 2021)”.
Line 319-322: Suggest the authors to separate this sentence as it is too long, and the points are not clear.
Line 323-326: It is difficult to follow and understand what did the authors want to express?

---

## Round 0.2 · Major Revisions

Thank you for your comprehensive revisions and detailed explanation of these. I appreciate the effort involved in these edits and your careful consideration of the suggestions that have been made.

You have addressed the majority of the reviewers concerns regarding the content and study design and/or provided justification for your methods. However, the manuscript requires further modification before it will be considered as suitable for publication in PeerJ.

I note that in your response to reviewers you have mentioned independent editing - however, the quality of the writing is not yet at the standard for PeerJ publication. The content does not follow a logical flow, there are numerous grammatical errors, and it is unclear (in parts) which relate to your study, and what is from other literature. Improving the presentation (and formatting) of the manuscript will help with the overall merit of the work, as at present, it is very confusing for a reader to follow. Please consider seeking further help with the style and writing of the manuscript.

Please also note that PeerJ requires exact p-values to be reported in the results section.

There are a few errors in the referencing and in-text citations - please check carefully throughout the manuscript regarding formatting of these.

Reviewer 2 ·

Basic reporting

Now reads well.

Experimental design

Now clearly explained

Validity of the findings

Good

---

## Round 0.3 · accepted · Accept

Thank you for your comprehensive revisions. We apologize for the delay in the decision.

The last version addressed the reviewers' extensive comments regarding the content, and this newly revised version addresses the comments regarding writing style and presentation of your data. This manuscript is ready for publication - congratulations.

There is one comment I would like you to consider: A reviewer has asked that you consider updating the title of your manuscript to "From the perspective of rumen microbiome and host metabolome, revealing the effects of feeding strategies on Jersey Cows on the Tibetan Plateau", as the term genomics is often used in a different context. Please consider if you would like to make this change when you are looking at your proofs.

Please make sure you carefully check the proofs at the next step.

Reviewer 3 ·

Basic reporting

The authors have addressed most of my questions and comments. Thanks for improving the manuscript!

Experimental design

The authors added the necessary information in the revised version.

Validity of the findings

no comment

Additional comments

I am still concerned about the title: "genomics" is more frequently used for the host. Therefore, I am wondering if the title should be changed to "From the perspective of rumen microbiome and host metabolome, revealing the effects of feeding strategies on Jersey Cows on the Tibetan Plateau".